# Robust Placeability Estimation for Model-Free Unified Pick-and-Place Reasoning

*Abstract*— **Reliable manipulation of previously unseen objects remains a fundamental challenge for autonomous robotic systems operating in unstructured environments. In particular, robust pick-and-place planning directly from noisy and only partial real-world observations, where object surfaces are inherently incomplete due to occlusions is difficult. As a result, many existing methods rely on strong object priors or assume placements on continuous, flat support surfaces such as planar tabletops, without explicitly accounting for edge proximity or inclined supports. In this work, we introduce a robust probabilistic placeability metric that evaluates 6D object placement poses from partial observations, allowing the generation of diverse multi-orientation placement candidates and condition grasp scoring on these placements, enabling model-free unified pick-and-place reasoning.**

## I. INTRODUCTION

The ability to pick objects and place them at desired locations is central to many robotic applications, including warehouse logistics, household assistance, and healthcare [1, 2]. In practice, achieving reliable pick-and-place execution in real-world environments requires perceiving an object's geometry and pose, planning stable grasps [3], identifying safe placements within target regions [4, 5], and executing these actions in confined spaces under uncertainty [6].

This interdependence has motivated recent work to reason jointly over grasping and placing, rather than treating them as independent problems [7–11], which we refer to as *unified pick-and-place reasoning*. When reasoning about the target placement happens already during grasp selection, robots can choose grasps that enable both successful picking and placement, while avoiding collisions early and reducing the need for re-grasping caused by environmental constraints. Yet, existing approaches primarily focus on tabletop-style planar supports, limiting placement reasoning in highly confined spaces. In particular, most similar to our work, Shanthi *et al.* [7] formulated pick-and-place as a constrained optimization problem that maximizes joint grasp and placement success probability. While effective, the method relies on accurate learned success models and can be sensitive to prediction errors.

Less explored in unified pick-and-place reasoning is how to incorporate prior work on object placement planning, which largely fall into two categories. First, classical *model-based placement* methods rely on geometric object models to evaluate placement stability, which are often not available in real-world scenarios [4, 5, 12–16]. Second, *model-free placement* methods can generalize to previously unseen objects from partial observations, but typically focus on predicting a single stable placing surface or pose in isolation, without

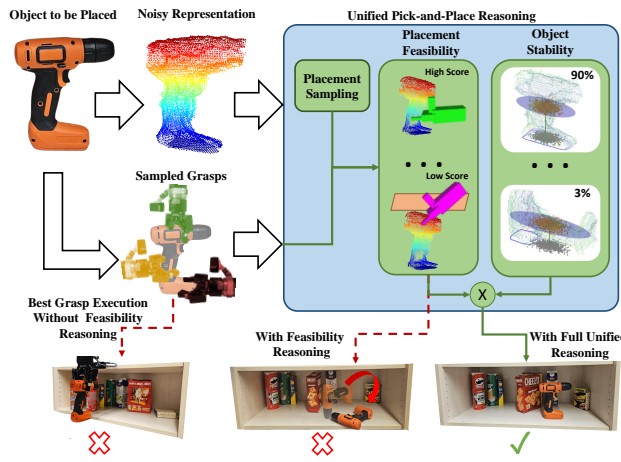

Fig. 1: From a noisy point cloud reconstruction (top left), candidate grasps and placement poses are jointly evaluated using estimated stability and placement feasibility (top right). The system selects the highest-scoring grasp-placement pair that is executable and estimated to be physically stable at the target location (bottom right). Note that selecting the grasp based only on grasp quality leads to a collision with the shelf (bottom left) and choosing a feasible grasp without stability reasoning can result in object tipping (bottom center).

adjusting to different support surfaces (often just assuming a table) and do not directly support task-level pick-and-place reasoning under robot and environment constraints [17–21].

Taken together, both directions leave an additional important gap, as they either lack physically grounded stability reasoning under realistic sensing noise or cannot robustly rank grasp-place pairs when object geometry and support conditions are only partially observed. This gap is often masked on tabletops but becomes a dominant failure mode in height-restricted shelves, where collision-free motion and object height severely restrict successful execution. In such settings, the robot must reason jointly about where an object can be stably placed and which grasps remain executable at the target location under collision and clearance constraints.

To address these limitations, we introduce a robust *placeability metric* that evaluates candidate 6D placement poses from partial point cloud observations by jointly scoring (i) probabilistic stability under reconstruction uncertainty and (ii) placement-conditioned graspability (PCG), ensuring that grasps remain reachable and collision-free at the placement pose. This enables efficient selection of stable and executable grasp-place pairs in confined environments as shown in Fig. 1. We demonstrate the effectiveness of our placeability metric within a model-free unified pick-and-place pipeline through a quantitative evaluation across varying scene configurations and objects.

## II. PROBLEM FORMULATION AND ASSUMPTIONS

We consider the following problem: Given source and target regions $S$ and $P$ reconstructed from sensor observations, and an object $o$, detected within $S$, with unknown 6D pose $m_o \in \mathrm{SE}(3)$, the task is to compute the best physically stable, reliably graspable, and collision-free placement pose $t_o$ out of a set of placement candidates $\mathcal{T}_P \in \mathrm{SE}(3)$ inside $P$ without a predefined position, orientation, and object model. The target area may exhibit cluttered objects and tight spatial constraints, which influence kinematic feasibility and collision-free placement. Therefore we reconstruct the workspace and the object online from RGB-D observations by fusing depth data into a truncated signed distance function (TSDF) scene mesh $\mathcal{M}$ [22].

## III. OUR APPROACH

To ensure robust object placement, we introduce an object-centric *placeability metric* that quantifies the physical feasibility of a candidate placement pose directly from the object's sensed geometry. The metric integrates two complementary components: (i) A **point cloud based probabilistic stability** measure that assigns higher scores to candidate placement poses estimated to remain in an static equilibrium. (ii) A **placement-conditioned graspability** measure that evaluates whether high-quality grasps remain kinematically feasible and collision-free after being transformed into a candidate placement pose, thereby linking grasp success directly to the following placement task. After introducing the individual components, we describe the unified grasp-place selection.

*1) Probabilistic Object Stability:* To find stable 6D placement poses for an object $o$, including placements on inclined surfaces or near environmental edges, we estimate the probability that the object remains in static equilibrium when placed at a candidate pose $t_o$ using a Monte Carlo sampling strategy. In particular, let $o_t$ be the object transformed to its pose at $t_o$, we then model the apriori unknown Center of Mass (CoM) as a Gaussian distribution $\mathcal{N}(\mu_{\mathrm{CoM}}, \Sigma_{\mathrm{CoM}})$, where the mean and covariance are computed from confidence-weighted points $p_i$ of the object's geometry $PC(o_t)$:

$$\mu_{\mathrm{CoM}} = \frac{\sum_i w_i p_i}{\sum_i w_i} \quad \Sigma_{\mathrm{CoM}} = \sum_i \left( \frac{w_i}{\sum_j w_j} \right)^2 \sigma_i^2 \quad (1)$$

Here, $\sigma_i^2$ is the estimated per-point variance and inversely proportional to the confidence values $w_i$ that are derived from TSDF weights, such that points with higher reconstruction confidence contribute more strongly to the CoM estimate.

To estimate the object's support regions which it can be safely placed on, we model potential contact points probabilistically and derive support polygons $SP(o_t)$ from these sampled contacts. First, we extract contact candidates $\mathcal{C}(o_t)$ of the object at its pose $t_o$

$$\begin{aligned} \mathcal{C}(o_t) &= \{ p_i \in PC(o_t) \mid z(p_i) \leq z_{\min} + \zeta \}, \\ z_{\min} &= \min_{p_j \in PC(o_t)} z(p_j), \end{aligned} \quad (2)$$

where $z(p_i)$ denotes the height of point $p_i$ and $\zeta$ is a small vertical tolerance. We then again assign each candidate a positional variance $\sigma_i^2$ derived from TSDF weights and convert these uncertainties into contact sampling probabilities using a softmax model. For each Monte Carlo sample $n = 1, \ldots, N$, we draw a subset $\mathcal{S}_n \subseteq \mathcal{C}(o_t)$ according to these probabilities and define a support polygon as the convex hull (CH) of their projections onto the support plane via a function $\Omega_{\mathrm{sp}}(\cdot)$:

$$SP_n(o_t) = \mathrm{CH}(\{\Omega_{\mathrm{sp}}(s) \mid s \in \mathcal{S}_n\}). \quad (3)$$

This procedure yields a distribution over support polygons $\{SP_n(o_t)\}_{n=1}^N$ that captures uncertainty in the contact configuration.

Finally, we evaluate overall object stability by drawing, for each Monte Carlo sample $n$, a CoM candidate $c_n$ from its distribution together with a support polygon $SP_n(o_t)$ from the probabilistic contact model of the object. We then project the CoM onto the support plane, and check whether $c_n$ lies inside the sampled support polygon. Based on that, the stability probability is defined as:

$$f_{\mathrm{st}}(o_t) = \frac{1}{N} \sum_{n=1}^N \left( \Omega_{\mathrm{sp}}(c_n) \in SP_n(o_t) \right) \quad (4)$$

The resulting stability value represents the probability that the object remains statically stable under reconstruction uncertainty and unknown internal mass distribution.

To improve robustness against execution errors and reconstruction noise of the target area, we additionally evaluate the stability under small random perturbations of the object pose. Specifically, we rotate the object point cloud $R$ times by randomly sampling pitch and roll offsets within $\pm 5°$ and compute the stability score for each perturbed pose. In this work we assume $R = 5$. The final probabilistic stability term is defined as the average over all trials:

$$\bar{f}_{\mathrm{st}}(o_t) = \frac{1}{R} \sum_{r=1}^R f_{\mathrm{st}}(o_t^r) \quad (5)$$

In summary, our formulation provides a probabilistic, uncertainty-aware stability estimate for arbitrary 6D object placement poses directly from reconstructed point clouds. This enables robust stability reasoning without requiring complete object models or deterministic contact assumptions.

*2) Placement-Conditioned Graspability (PCG):* To validate whether a candidate placement pose can actually be realized, we introduce placement-conditioned graspability, which evaluates if a grasp remains feasible once transformed into the placing pose frame. Let $\mathcal{G}(o) = \{g_0, \ldots, g_K\}$ be the set of $K$ candidate grasps for object $o$ at the object's observed pose $m_o$. Then, $T(t_o)$ denotes the rigid transform of a grasp $g_k$ from its sampled grasp pose at $m_o$ to a placement pose $t_o$. The resulting transformed grasp, which we refer to as *place-grasp*, is defined as $g_k^{t_o} = T(t_o) \cdot g_k$.

To compute the PCG, we score a grasp candidate $g_k$ against a possible placement candidate $t_o \in \mathcal{T}_P$ to check if it remains reachable and collision-free:

$$f_{\mathrm{pcg}}(g_k^{t_o}) = f_{\mathrm{RM}}(g_k^{t_o}) \cdot f_{\mathrm{coll}}(g_k^{t_o}, \mathcal{M}) \cdot f_{\mathrm{alt}}(g_k^{t_o}) \quad (6)$$

where $f_{\mathrm{RM}}(\cdot) \in \{0, 1\}$ tests kinematic feasibility for the manipulator via a precomputed reachability map [23] and $f_{\mathrm{coll}}(\cdot, \mathcal{M}) \in \{0, 1\}$ checks a simplified gripper model against the environment mesh $\mathcal{M}$ of the target region for collision. Lastly, to avoid collisions with the environmental supporting surface, we enforce a minimum vertical clearance ($\delta_{\min}$) between a candidate grasp (with vertical height $z(g_k^{t_o})$) and the object's lowest sensed point ($z_{\min}$), with:

$$f_{\mathrm{alt}}(g_k^{t_o}) = \begin{cases} 1, & \text{if } z(g_k^{t_o}) - z_{\min}. \geq \delta_{\min}, \\ 0, & \text{otherwise.} \end{cases} \quad (7)$$

This formulation ensures that placement candidates are evaluated only based on grasps that remain kinematically feasible and collision-free at the target pose.

### A. Unified Pick-and-Place Reasoning

After introducing the components of our placeability metric, we now present our approach for model-free unified pick-and-place reasoning, which integrates grasp candidate selection and placement evaluation into a single pipeline.

*1) Placement Candidate Sampling:* First, we compute the set of all possible placement candidates $\mathcal{T}_P$ on the simplified triangle mesh $\mathcal{M}$ of $P$ by extracting horizontal surfaces and sampling a fixed set of points [24]. Furthermore, to increase orientation diversity among the placement poses, we apply a random $yaw$, as well as rotations around $\pm 90°$ and $180°$ $pitch$ and $roll$ axes, resulting in a total of six candidate orientations.

*2) Graspability Scoring:* For grasp candidate generation on object point clouds, we use the Grasp Pose Detection (GPD) network [25] and retain only the top-$k$ grasps $\{g_1, \ldots, g_k\} \subseteq \mathcal{G}(o)$ based on their predicted grasp scores $q_g(g)$. Note that any grasp prediction method can be used, provided that grasps are represented in SE(3).

*3) Placeability Scoring:* Finally, we define placeability for each candidate place-grasp $g_k$ of an object $o_t$ transformed to a candidate placement pose $t_o$ by combining probabilistic stability and grasp feasibility. The overall placeability score is then given by

$$q_t(g_k, \mathcal{T}_P) = \frac{1}{|\mathcal{T}_P|} \sum_{t_o \in \mathcal{T}_P} \bar{f}_{\mathrm{st}}(o_t) \cdot f_{\mathrm{pcg}}(g_k^{t_o}) \quad (8)$$

To rank grasps according to both pick and place feasibility, we define a unified grasp score that considers jointly the original grasp quality $q_g(g_k)$ as well as the associated stability and feasibility of the possible placements:

$$q_{gt}(g_k, \mathcal{T}_P) = q_g(g_k) \cdot q_t(g_k, \mathcal{T}_P) \quad (9)$$

This formulation inherently favors grasps that exhibit high intrinsic quality and are associated with stable, accessible placements (as shown in Figure 2). Finally, we execute the feasible place-grasp with the highest score according to $q_{gt}$, together with the most stable placement attainable with that grasp.

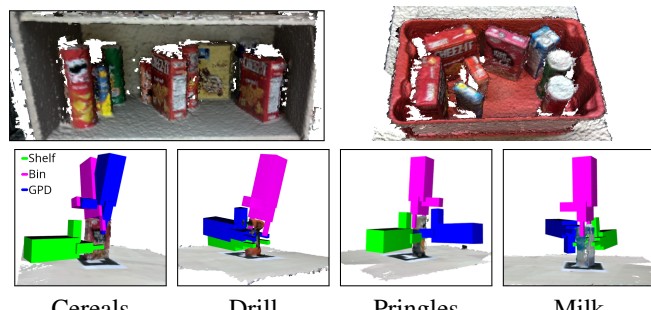

| Cereals | Drill | Pringles | Milk |

Fig. 2: Qualitative grasp comparison for two placement scenes: a shelf (top left) and a bin (top right). Top-ranked grasp candidates are shown for four perceived objects. Blue indicates the original GPD ranking [25], while green (shelf) and magenta (bin) indicate the grasps re-ranked using our placeability metric.

## IV. EXPERIMENTAL EVALUATION

To evaluate the effectiveness of our placeability-informed unified pick-and-place pipeline (**UniP**), we conduct quantitative real-world experiments on a UR5e platform and compare against two sequential baselines and one ablated variant[1]. Each method is tested with three pick-and-place trials per object in two shelf configurations (Fig. 3). The first scenario considers a cluttered shelf without severe spatial constraints. The second has strongly reduced vertical clearance, making grasp and placement selection more restrictive.

*1) Performance Comparison with Baselines:* The first baseline, **Grasp-RP**, follows a sequential pick-then-place strategy: it selects the highest-scoring grasp predicted by GPD [25] and assigns a random placement pose from $\mathcal{T}_P$ (see Sec. III-A.1) while preserving the object's original orientation. **Grasp-MO** uses the same sequential strategy but allows six object orientations, isolating the effect of multi-orientation reasoning without explicit feasibility evaluation. **UniP-NoStab** is an ablated version of our unified pipeline without the probabilistic stability term. We evaluate all methods by success rate and failure type (Table I).

In the cluttered environment, our **UniP** achieves the highest success rate 93.4% with only a single failure caused by grasp execution. In contrast, both sequential baselines (**Grasp-RP** and **Grasp-MO**) perform substantially worse. Although allowing multiple object orientations increases placement opportunities, frequent failures due to infeasible place-grasps and unstable configurations remain. The unified ablation **UniP-NoStab** achieves 86.8%, showing that jointly reasoning about grasp and placement feasibility already provides a substantial improvement. The advantage of our full method becomes especially present in the height-reduced environment, where **UniP** maintains high success, while the sequential baselines degrade significantly and **UniP-NoStab** drops in success primarily due to unstable placements.

Overall, these results confirm that the components of our method are necessary, i.e., *stability reasoning prevents physically invalid placements, while unified grasp–placement feasibility reduces execution failures.*

---

[1]Video: https://bit.ly/ICRAWS

| | Cluttered Shelf Environment | | | | Height-Reduced Environment | | | |
|---|---|---|---|---|---|---|---|---|
| Object | Grasp-RP | Grasp-MO | UniP-NoStab | **UniP (Ours)** | Grasp-RP | Grasp-MO | UniP-NoStab | **UniP (Ours)** |
| Drill | 0% | 0% | 67% | **100%** | 33% | 0% | 33% | **100%** |
| Cereals | 33% | 100% | 100% | **100%** | 0% | 0% | 100% | **100%** |
| Pringles | 67% | 100% | 100% | **100%** | 0% | 0% | 67% | **67%** |
| Mustard | 100% | 67% | 67% | **100%** | 67% | 100% | 33% | **67%** |
| Milk | 33% | 33% | 100% | **67%** | 33% | 0% | 67% | **100%** |
| Average | 46.6% | 60.0% | 86.8% | **93.4%** | 26.6% | 20% | 60.0% | **86.8%** |
| Failure Type | Failure Breakdown | | | | | | | |
| Grasp failed, place found | 2 | – | – | 1 | 2 | 3 | 2 | 2 |
| Grasp valid, No place found | 6 | 4 | – | – | 9 | 8 | – | – |
| Unstable placing | – | 1 | 2 | – | – | 1 | 4 | – |

TABLE I: Real-world pick-and-place success rates across five objects in two shelf configurations. UniP achieves the best overall performance, especially under reduced vertical clearance where the sequential baselines Grasp-RP and Grasp-MO degrade substantially, while UniP-NoStab, despite outperforming them, remains inferior to UniP due to unstable placements.

Cluttered Shelf     Height-Reduced Shelf

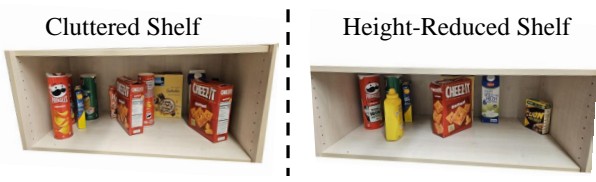

Fig. 3: Illustration of our two evaluation scenes: a cluttered shelf environment, and a height-reduced environment with strongly reduced vertical clearance.

*2) Runtime Analysis:* To assess the online performance of our full pick-and-place reasoning pipeline, we measured the execution time across all objects and trials in the cluttered shelf, resulting in an average runtime of $183 \pm 21$ s. The majority of this time ($178 \pm 21$ s) was spent on robot motion planning, motion execution, grasp computation, and reconstruction. In contrast, evaluating our placeability metric, including probabilistic stability and placement feasibility, required only $5 \pm 0.3$ s, indicating that it can be computed online with negligible impact on the overall task time.

*3) Place-Grasp Evaluation:* To evaluate the adaptability of our unified reasoning approach, we assess how grasp selection changes when explicitly conditioned on different target placement regions. Therefore, we run the full pipeline on four objects in those two scenarios. The results, shown in Fig. 2, demonstrate that our placeability-based rescoring systematically adapts grasp selection to the placement context without requiring any parameter changes. While GPD does not adapt grasp selection to the target region and may select grasps that are feasible for one placement area but not for another, our proposed metric adapts grasp selection when necessary while preserving high-quality grasps that remain executable at the target placement.

*4) Stability estimation:* Beyond planar placements, a stability metric must predict static equilibrium on arbitrary support surfaces. We evaluate our formulation (Sec. III-.1) in two settings: edge proximity and surface inclination.

As a baseline, we compute tipping thresholds using the CAD model's center of mass, following prior work that approximates the CoM from object geometry alone [5]. For edge proximity, we obtained real-world tipping thresholds by gradually translating each object toward a support boundary until static equilibrium was lost, and report the threshold as the percentage of the object's support footprint remaining in contact with the surface. As shown in Fig. 4.a–b, our stability estimates exhibit a transition near the measured real-world

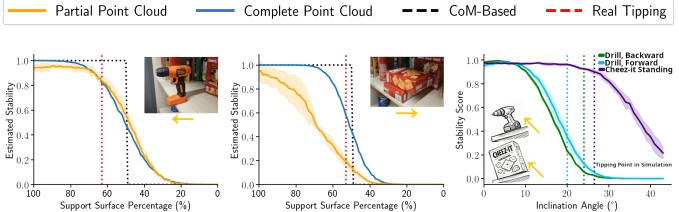

(a) Drill Forward     (b) Cheez-It Box     (c) Surface Inclination

Fig. 4: Edge-proximity and inclination tipping analysis showing the predicted probabilistic stability score while translating the object toward a support surface edge (a)–(b) and increasing the environmental support plane inclination (c). Blue curves indicate evaluation using the complete ground-truth mesh point cloud, while all remaining curves use partial noisy point clouds.

tipping thresholds across all objects. In contrast, the CAD-based CoM baseline deviates near the threshold, as it ignores contact geometry and partial support effects.

For potential object stacking cases, we evaluate stability under increasing support tilt in simulation to obtain repeatable tipping angles. In particular, we initialize the object in a stable pose on a planar support and increment the support inclination until static equilibrium is lost. As shown in Fig. 4.c, the predicted stability decreases monotonically with tilt and drops near the simulated tipping angle for both drill orientations, whereas for the Cheez-It box it remains overly optimistic until shortly before toppling, likely due to the absence of friction modeling. Nevertheless, these results demonstrate that *our proposed metric reliably captures geometry- and center-of-mass–driven tipping behavior*.

## V. Conclusion

In this work, we introduced a probabilistic placeability metric for model-free unified pick-and-place reasoning from noisy partial point clouds. By jointly evaluating probabilistic stability, and placement-conditioned graspability, the proposed method selects grasp–placement pairs that are both executable and physically stable without requiring CAD models or predefined placement poses. Experiments show that our stability formulation predicts tipping behavior under edge proximity and surface inclination directly from partial observations. Further real-robot experiments demonstrate that the full pipeline outperforms sequential baselines and an ablation without stability reasoning, highlighting the importance of both unified grasp-placement reasoning and probabilistic stability estimation.

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
