# OpenReview forum: "Robust Placeability Estimation for Model-Free Unified Pick-and-Place Reasoning"
_IEEE.org/ICRA/2026/Workshop/Manipulation_Robustness — ICRA 2026_

### Official Review · Reviewer_sbcr · 2026-05-06
**Good paper in line with the workshop scope**

**Rating:** 8
**Confidence:** 4

**Review:**

Reviewer 1:
Strengths:
1. The paper addresses an important and relatively under-explored problem: conditioning grasp selection on downstream placement feasibility and stability. 2. The proposed placeability metric is intuitive and practically motivated, combining probabilistic stability estimation with placement-conditioned graspability. 3. The real-robot experiments show that the method can effectively improve pick-and-place success over sequential baselines and the no-stability ablation.

Weaknesses:
1. The experimental evaluation is limited, with only a small number of objects, shelf configurations, and trials. 2. The method may also be computationally expensive, since it requires multiple placement candidates, Monte Carlo stability sampling, perturbation evaluation, and reachability/collision checks. 3. The method assumes that the target placement region and candidate final placement poses are known before grasp execution, which may limit its applicability to more open-ended manipulation tasks.

Overall:
This is a good workshop paper with a clear problem motivation and promising experimental results. I recommend acceptance, while encouraging the authors to expand the experiments and further discuss computational cost and placement-pose assumptions.
////////////////
Reviewer 2:
This work presents a probabilistic estimation metric that incorporates grasp candidate scoring and placeability evaluation, enabling robust and safe pick-and-place task executions. The authors demonstrate the approach's strength in cluttered and low-height shelf placing pick-and-place tasks, and study edge-proximity stability estimation of various objects. The paper is well written and structured.
While the idea of the probabilistic object stability metric is well-explained, an additional visualization of the generation of supporting polygons and contact configuration would help the reader to better imagine the derivation.
The experiments show that UniP outperforms all baselines, but it is left unclear why it failed in the height-reduced setup and if the computation time is significantly different from the cluttered shelf case.
In IV Experimental Evaluation - Section 4) it is claimed that the “stability estimation exhibits a transition near the measured real-world tipping thresholds”, which contradicts Fig 4 (a), where a stability of more than 80% is predicted for the drill at ~65% support surface despite the object tipping over in real world. Here, a more detailed analysis would be interesting.
At last, the number of experiment setups could be increased e.g. pick-and-place tasks where the optimal grasping pose is unreachable or objects exceeding the shelf height, such as a large cereal box that can only be placed horizontally in the shelf.

---

### Decision · Program_Chairs · 2026-05-21

Accept